# Epidemiology of patients treated for multiple myeloma using a new algorithm in the French national health insurance database (SNDS): Results from the MYLORD study

Cyrille Touzeau[1]*, Fanny Raguideau[2], Hélène Denis[2], Ludovic Lamarsalle[2], Caroline Guilmet[3], Matthieu Javelot[3], Marie Pierres[3], Aurore Perrot[4]

1 Centre Hospitalier Universitaire de Nantes, Nantes, France, 2 HEVA, Lyon, France, 3 Janssen-cilag France, Issy-les-Moulineaux, France, 4 Institut Universitaire du Cancer Toulouse – Oncopole, Toulouse, France

* Cyrille.Touzeau@chu-nantes.fr

## Abstract

### Background

Management of multiple myeloma (MM) has evolved in the last decade, impacting epidemiology and creating the need for a revised algorithm for patient identification in the French National Health Insurance database (SNDS). The objective of the MYLORD study was to provide up-to-date epidemiological indicators for MM using the SNDS.

### Methods

Adults treated for MM between 2014 and 2020 were identified using hospital stays (ICD-10 code C90*) and/or Long-Term Disease status for MM (ICD-10 C90*) and treatment dispensation (MM specific treatments), or lenalidomide or thalidomide with 2 conditions: at least 2 Serum or Urine-Protein Electrophoresis performed within 4 months after the first drug dispensing and no hospital stays for other indications than MM.

### Results

In 2020 in France, the number of new patients treated for MM was estimated at 5,608, corresponding to a world age-standardized incidence rate (ASIR) of 3.67 (3.55–3.78) per 100,000 patients. The total number of patients treated for MM was estimated at 33,675 corresponding to a world age-standardized prevalence rate (ASPR) of 22.34 (22.06–22.63) per 100,000 patients. The world ASIR slightly increased from 2014 to 2018, then slightly decreased until 2020. There is a regular increase of the ASPR every year in both men and women.

**Data availability statement:** The data supporting the study findings are part of the National health data system (SNDS, Système national des Données de Santé) and are available from the Health Data Hub (HDH https://www. health-data-hub.fr/). Restrictions apply to the availability of these data and the code used for their analysis, which were used with a special permission for this study.

**Funding:** This study was funded by Janssen Cilag France. The funders had no role in study design, data collection and analysis, decision to publish, or preparation of the manuscript.

**Competing interests:** I have read the journal's policy and the authors of this manuscript have the following competing interests: CT/AP: JANSSEN, Amgen, BMS/Celgene, GSK, Takeda and Sanofi. MJ are MP are employees of Janssen Cilag France. CG was also an employee of Janssen Cilag France at the time this study was conducted. FR, LL, HD are employees of HEVA, who received financial compensation from JANSSEN Cilag France to conduct the study. his does not alter our adherence to PLOS ONE policies on sharing data and materials.

## Conclusions

The algorithm developed for the MYLORD study considerably improves the identification of patients treated for MM in the SNDS compared to the current French references and allows to generate robust epidemiological indicators. The stable incidence and the increase in prevalence from 2014 to 2020 reflect the improvement in therapeutic management over the years.

## Introduction

Multiple myeloma (MM) is a bone marrow cancer involving plasma cells mostly affecting the elderly [1]. Symptoms include skeletal pain and fatigue which impact the patient's quality of life, as well as evidence of end-organ damage (the so-called CRAB criteria: hypercalcemia, renal insufficiency, anemia and/or bone lesions) [2,3].

In 2019, there were 155,688 MM cases worldwide, with an age-standardized incidence rate (ASIR) of 1.92/100,000 persons [4]. The incidence of MM increases with age. In metropolitan France, MM/plasmacytoma was one of the leading malignant hemopathy in 2018, with 5,442 new cases estimated. Median age at diagnosis was 70 years old in men and 74 years old in women [5]. MM is currently an incurable disease, but the fast therapeutic advances of recent years have had a positive impact on patient survival [6,7].

There are 2 main information systems that collect data on MM patients in France. The first one is the French network of cancer registries (FRANCIM) that covers 20% of the national population. The latest published figures are based on data collected between 1990 and 2015, extrapolated to metropolitan France level and projected for 2016–2018. This lag time implies that the figures do not reflect the latest treatment evolutions and their impact on the disease epidemiology. The second one is the French National Health database (SNDS) that collects nearly all health-care consumption of the entire French population. To date, PALMARO is the only published algorithm for MM patients' identification [8] based on LTD (Long-Term Disease) status and hospitalization. This algorithm was validated using a regional (Tarn) center cancer registry as the standard reference [8,9]. With a sensitivity of 90%, a specificity of 100% and a positive predictive value of 60% it failed to consider the recent evolution of the MM treatment landscape.

Indeed, the management of patients with MM has evolved. Patients are increasingly being managed in the outpatient setting (including per os treatment delivered in retail pharmacy); therefore, identification algorithms centered on hospitalizations are no longer efficient. Moreover, a feasibility analysis of the SNDS dataset showed limited reporting of LTD in MM patients. We hypothesized that the incidence and prevalence of MM in France had been underestimated by the PALMARO algorithm. We propose to take into account the change of treatment management to improve the identification of MM in the SNDS database.

This work provides up-to-date incidence, prevalence, and mortality figures using an improved algorithm and compares these epidemiological indicators to those obtained with the 2 above mentioned methods as a validation mechanism.

## Materials and methods

### Setting and design

MYLORD study is a retrospective cohort study of adults treated for MM, identified through the SNDS from 2006 to 2020 and alive on 1 January 2014.

### Data sources

The study was performed using the SNDS that collects individual data used for billing and reimbursement of outpatient healthcare linked to the National Hospital Discharge database (PMSI) by means of a unique anonymous identifier allocated to each individual, the social security number [10]. The SNDS covers almost the entire French population from birth (or immigration) to death (or emigration). The reason for hospitalization (principal diagnosis (PD) and/or related diagnosis (RD) and/or significant associated diagnosis (SAD)) can be retrieved from the anonymized hospitalization summary. In the SNDS, most of treatments for MM patients are recorded (excepted those from clinical trials and intra-Diagnostic Related Groups (DRG list treatments)). LTD status is available and coded according to the ICD-10. This special status grants full coverage of medical expenses. The SNDS comprises several insurance schemes. Individuals and their families are covered by a scheme based on their employment status and remain covered by this scheme after retirement. Of the 66 million inhabitants in France at the end of 2015, the general scheme (excluding the local mutualized healthcare sections [SLM]) covers 76% of the population living in France, including private and public employees, students, and unemployed people [10].

The MYLORD study included only patients affiliated to the general scheme because vital status is accurately reported in the SNDS only for patients affiliated with this scheme [10].

### Study population–Identification of patients treated for MM (MYLORD algorithm)

All adults (≥18 years) with at least one hospital stay for MM (PD, RD or SAD with ICD-10 C90*) and/or with LTD status for MM (ICD-10 C90*) and treated by MM specific drugs or chemotherapy or Autologous Stem Cell Transplantation (ASCT) from January 1st, 2006, to December 31st, 2020 were included (treatment list in S1 Appendix in S1 File). In addition, in the same period, all adults treated by lenalidomide or thalidomide (the same drug dispensed at least twice within 90 days) AND with at least 2 serum protein electrophoresis (SPEP)/ urine protein electrophoresis (UPEP)within 4 months of the year after first dispensing of the drug and without hospital records on the database for indications other than MM (diagnosis codes list in S1 Appendix in S1 File) were included. Other indications were: myelodysplastic syndrome and/or follicular lymphoma and/or diffuse non-Hodgkin lymphoma and/or peripheral and cutaneous T cell lymphoma and/or other non-Hodgkin lymphoma and/or osteomyelofibrosis and/or acute panmyelosis with myelofibrosis and/or POEMS syndrome and/or amyloidosis.

The two SPEP/UPEP (associated with checking for hospitalization diagnosis) were added to the inclusion criteria to ensure that only MM patients were selected, as lenalidomide and thalidomide are also used to treat other diseases.

We restricted the MYLORD algorithm to only treated patients to exclude smoldering (or indolent) MM (SMM).

### Study periods

The study period was from January 1st, 2014 to December 31th, 2020. A historical period was available since January 1st, 2006. The index date was defined as the first date with MM treatment during the study period (2014–2020) for patients without MM treatment before January 1st, 2014 (incident patients). The index date was January 1st, 2014 for patients with MM treatment before January 1st, 2014 (prevalent patients). Patients were followed from first date with MM information (hospitalization, LTD status or treatment) until death or until the end of study or until a period of 2 consecutive years without any reimbursement, whichever occurred first.

## Statistical analyses

Descriptive statistics were used to characterize the study population. Outcomes were assessed by calendar year constituting yearly sub-populations. Prevalence was defined as all patients eligible in the study who were alive and affiliated to the general scheme on January 1st of the considered year. Incidence was defined as all patients without MM information for 2 years prior to their inclusion. Incident patients were all newly included patients on the considered year of analysis. Patients could be considered incident only once during the study period. Crude rates were defined as the number of patients (prevalent, incident, or dead) divided by the number of people affiliated to the General Scheme the corresponding year. All rates were standardized using the age distribution in the worldwide population provided age-standardized incidence rate (ASIR) and age standardized prevalence rate (ASPR) [11]. Results were extrapolated to metropolitan and overseas (i.e., French territories outside Europe) France and are available in S3 and S5 Tables in S1 File. Statistical analyses were performed using SAS version 9.4.

## External validation of the MYLORD algorithm using the PALMARO et al. algorithm

We applied a previously published algorithm of the identification of MM patients in the SNDS and we compared the estimated indicators to those obtained in MYLORD study [8].

As suggested by the Palmaro et al. algorithm, all adults (>18 years) with at least one hospital stay for MM (PD, RD or SAD with ICD-10 C90*) and/or with LTD status for MM (ICD-10 C90*) from January 1st, 2014 to December 31st, 2020 were included, regardless of treatment status [8]. Incident patients were patients without MM information within two years prior the first date of MM information during the study period. To be comparable to the MYLORD algorithm, we tested two other alternatives: one with an identification period starting from 2006 and one keeping the same identification period but including only treated patients (using the same list of treatments as MYLORD).

For comparison purposes, results obtained with the MYLORD and the PALMARO algorithms in the general scheme population were extrapolated to the metropolitan French population using the number of prevalent (and incident) patients standardized by the age and sex distribution in metropolitan France in the corresponding year provided by the INSEE (*Institut national de la statistique et des études économiques*).

## Ethics statement

The SNDS is a medico-administrative database of insured persons, and all patient-level data used for this retrospective analysis were collected as part of routine diagnosis and treatment. A unique anonymous identification number was associated to each insured person; as such, all data were fully anonymized prior to access and inclusion in this study.

In accordance with the regulations in force, patient consent was not necessary because this study uses secondary data, there was a public interest in assessing the MM population in France and its therapeutic management, and the protection of patients' rights and freedom were guaranteed. The authorization to use the data was granted by the French data protection authority (*Commission Nationale de l'Informatique et des Libertés,* CNIL) (Decision DR-2019–103, and authorization No. 918419) and the study protocol obtained approval from the committee for research, studies, and evaluations in the field of health (*Comité d'expertise pour les recherches, les études et les évaluations dans le domaine de la santé*, CEREES) (Decision TPS 204907). The initial convention to access the SNDS data was signed on July 2019 with the French National Health Insurance (*Caisse Nationale Assurance Maladie,* CNAM), and the last dataset was received on March 31st, 2023.

## Results

### Study population

**MYLORD.** Using the MYLORD algorithm, we identified 46,202 MM-treated adults from 2014 to 2020 (Flow chart in S1 Fig in S1 File). Among them, a hospital stay for MM was the most frequent first MM information reported during

the inclusion period (N = 29,591; 64.1%). Moreover, for more than a quarter of the patients (N = 11,838; 25.6%) the first reported MM information was the combination of all-oral treatment regimens with thalidomide or lenalidomide (at least twice) and 2 SPEP/UPEP within 4 months of the year after first dispensing of the drug without hospital records on the database for indications other than MM (diagnosis codes list in S1 Appendix in S1 File). This proportion increased over the years: in 2014, it represented 21.2% of patients, whereas it increased to 30.5% in 2020.

**PALMARO.** With the PALMARO algorithm, we identified 59,704 MM adults and 36,453 MM-treated adults from 2014 to 2020. Among the 59,704 MM adults, a LTD record was the most frequent first MM information reported during the inclusion period (N = 31,214; 52.3%). A hospital stay was initially recorded for 43.8% of the patients (N = 26,131). The remaining patients (N = 2,359; 3.9%) had both hospital stay and LTD reported the same day.

The extension of the identification period to 2006 permitted identification of some additional patients increasing the number of MM adults to 62,688 and the number of MM-treated adults to 36,674.

## Study population characteristics

**MYLORD.** Patients treated for MM between 2014 and 2020 had a mean age (SD) of 69.5 years (12.9) and a median age of 71.0 years (Table 1). There were 50.2% men (n = 23,212) and 49.8% women (n = 22,990). There was no notable evolution in median age (S1 Table in S1 File). The gender distribution tended to also remain stable over the years. About 61% of the patients had the LTD status for MM registered in the databases in 2020.

**PALMARO.** Patients with MM had a mean age (SD) of 71.8 (12.5) and a median age of 73.0 years (Table 1). There were 50.3% men (n = 30,007) and 49.7% women (n = 29,697). MM-treated adults had a mean age (SD) of 70.3 (11.6) and a median age of 71 years. There were 51.2% men (n = 18,675) and 48.8% women (n = 17,778). Patients identified with the PALMARO algorithm (regardless of treatment status) tended to be a little older than those identified with the MYLORD algorithm, whereas the sex ratio was similar. Patients treated for MM identified with the PALMARO algorithm had a similar median age and sex ratio to those identified with the MYLORD algorithm.

## Epidemiological indicators

### Based on the MYLORD algorithm.

- Incidence (Table 2, Figs 1 and 2)

In 2020, the number of new patients treated for myeloma in metropolitan France was estimated at 5,402 (Fig 1), corresponding to a world ASIR of 3.67 (3.55–3.78) for 100,000 patients (S4 Table in S1 File). This incidence rate remained a little higher in men than women. Indeed, in 2020, the world ASIR rate of MM was 4.12 (3.94–4.30) in men and 3.29 (3.14–3.45) in women. These rates tended to slightly increase from 2014 to 2018 and then slightly decreased until 2020 (Fig 2).

**Table 1. Comparison of characteristics at inclusion of patients identified with the MYLORD and the PALMARO algorithms among patients affiliated to the health insurance's general scheme in France (between 2014-2020).**

|  |  | MYLORD N = 46,202 | PALMARO (regardless of treatment status) N = 59,704 | PALMARO (treated patients) N = 36,453 |
|---|---|---|---|---|
| Age (in years) | Mean (± SD) | 69.54 (+/-12.92) | 71.76 (+/-12.50) | 70.27 (+/-11.64) |
|  | Median | 71.00 | 73.00 | 71.00 |
|  | Q1; Q3 | 62.00; 79.00 | 64.00; 81.00 | 63.00; 79.00 |
| Sex, n (%) | Men | 23,212 (50.24%) | 30,007 (50.26%) | 18,675 (51.23%) |
|  | Women | 22,990 (49.76%) | 29,697 (49.74%) | 17,778 (48.77%) |

**Table 2.** Incidence of MM patients in metropolitan France: comparison of the MYLORD, the PALMARO and the FRANCIM registries figures (2014–2020).

| | Year | MYLORD | PALMARO Regardless of treatment status | PALMARO Treated only | FRANCIM registries[1] |
|---|---|---|---|---|---|
| Number of incident patients | 2014 | 4,985 | 6,166 | 4,264 | 5,049 |
| | 2015 | 5,016 | 6,035 | 4,204 | 5,154 |
| | 2016 | 5,351 | 6,295 | 4,307 | 5,260 |
| | 2017 | 5,642 | 6,556 | 4,522 | 5,370 |
| | 2018 | 5,665 | 6,270 | 4,406 | 5,442 |
| | 2019 | 5,550 | 6,266 | 4,259 | Not available |
| | 2020 | 5,402 | 6,071 | 3,699 | Not available |
| Standardized incidence rate for 100,000 PY -Men (95% CI) | 2014 | 4.24 (4.05-4.44) | 5.19 (4.98-5.41) | 3.72 (3.54-3.90) | 4,1 |
| | 2015 | 4.25 (4.05-4.45) | 4.90 (4.70-5.11) | 3.49 (3.31-3.67) | 4,1 |
| | 2016 | 4.37 (4.17-4.57) | 5.28 (5.07-5.49) | 3.69 (3.51-3.87) | 4,1 |
| | 2017 | 4.74 (4.54-4.94) | 5.42 (5.21-5.63) | 3.89 (3.71-4.07) | 4,1 |
| | 2018 | 4.77 (4.57-4.97) | 5.15 (4.95-5.36) | 3.77 (3.59-3.95) | 4,2 (3.9-4.5) |
| | 2019 | 4.37 (4.18-4.55) | 4.95 (4.75-5.14) | 3.47 (3.31-3.64) | Not available |
| | 2020 | 4.12 (3.94-4.30) | 4.72 (4.53-4.91) | 2.88 (2.73-3.03) | Not available |
| Standardized incidence rate for 100,000 PY -Women (95% CI) | 2014 | 3.43 (3.26-3.60) | 3.76 (3.59-3.93) | 2.71 (2.57-2.86) | 2,9 |
| | 2015 | 3.42 (3.25-3.59) | 3.76 (3.59-3.92) | 2.75 (2.61-2.90) | 2,9 |
| | 2016 | 3.57 (3.40-3.74) | 3.67 (3.50-3.84) | 2.62 (2.47-2.76) | 2,9 |
| | 2017 | 3.51 (3.34-3.68) | 3.77 (3.60-3.94) | 2.63 (2.49-2.77) | 2,9 |
| | 2018 | 3.34 (3.18-3.50) | 3.53 (3.37-3.69) | 2.50 (2.37-2.64) | 2,9 (2.7-3.2) |
| | 2019 | 3.33 (3.17-3.49) | 3.53 (3.38-3.69) | 2.38 (2.25-2.51) | Not available |
| | 2020 | 3.29 (3.14-3.45) | 3.42 (3.27-3.57) | 2.12 (2.00-2.24) | Not available |

[1]Source: https://www.santepubliquefrance.fr/docs/estimations-nationales-de-l-incidence-et-de-la-mortalite-par-cancer-en-france-metropolitaine-entre-1990-et-2018-volume-2-hemopathies-malignes.

- Prevalence (Table 3, Figs 1 and 2)

    In 2020, the total number of patients treated for myeloma in metropolitan France was estimated at 32,491(Fig 1), corresponding to a world ASPR of 22.34 (22.06–22.63) for 100,000 patients (S2 Table in S1 File). ASPR remain a little higher in men than in women. Indeed, in 2020, the world ASPR for 100,000 patients was 24.48 (24.04–24.91) in men and 20.62 (20.24–21.01) in women. There is a slight regular increase of the ASPR annually in both men and women (Fig 2).

- Mortality (Table 4)

    In 2020, the MM world age-standardized mortality rate for 100,000 patients was of 1.97 (1.90–2.05) (Table 4). This rate remained relatively stable over time.

    **Comparison of the MYLORD, PALMARO, and FRANCIM epidemiological indicators (Table 2).** We used the year 2018 for comparison purposes as it was the latest available year for FRANCIM published data. The number of incident

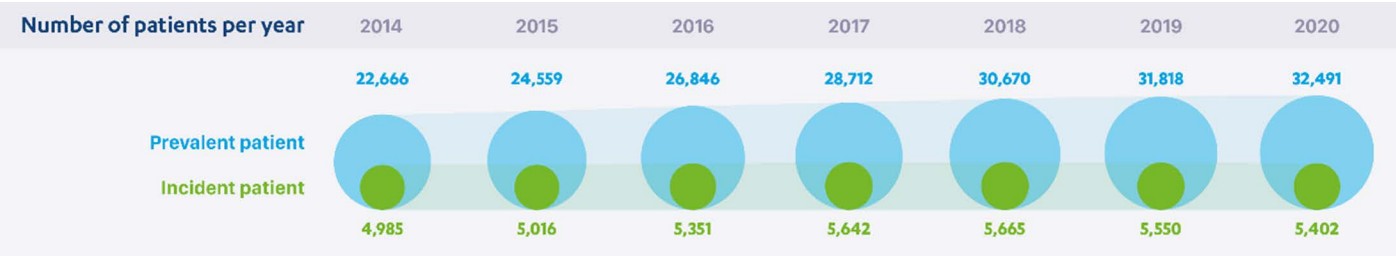

**Fig 1. Number of MM patients in metropolitan France (2014-2020).**

patients treated for myeloma extrapolated to metropolitan France was 5,665 (3,064 men and 2,601 women) with the MYLORD algorithm. This number increased to 6,270 adults when applying the PALMARO algorithm but decreased to 4,406 adults when we restricted this algorithm to treated patients. The FRANCIM estimation is closer to MYLORD than PALMARO, with 5,442 incident MM patients (2,822 men and 2,620 women). The world ASIR in men for 100,000 patients was 4.77 (4.57–4.97) in MYLORD, 5.15 (4.95–5.36) with PALMARO (3.77 (3.59–3.95) when PALMARO data was restricted to treated patients) and 4.2 (3.9–4.5) for FRANCIM. The world ASPR in women for 100,000 patients was 3.34 (3.18–3.50) in MYLORD, 3.53 (3.37–3.69) with PALMARO (2.50 (2.37–2.64) when PALMARO data was restricted to treated patients) and 2.9 (2.7–3.2) for FRANCIM. Incidence increased over time from 2014 to 2018 with MYLORD, PALMARO and FRANCIM registries. We observed a slightly decreased incidence in 2019–2020 with MYLORD and PALMARO algorithms. No data are available for FRANCIM for 2019–2020.

## Discussion

For the 2014–2020 period, we identified 46,202 patients treated for MM with the MYLORD algorithm, that is 9,749 more patients than would have been identified with the PALMARO algorithm for treated patients.

Incidence rates tended to slightly increase from 2014 to 2018 and then slightly decrease until 2020 with around 4.1 new cases per 100,000 patient years (PY) in men and 3.3 new cases per 100,000 PY in women. The slight decrease observed at the end of the study period is probably due to a delay in diagnosis because of the COVID-19 pandemic. Prevalence rates increased regularly from 2014 to 2020 in both men and women with around 24.5 cases per 100,000 PY in men and 20.6 cases per 100,000 PY in women in 2020. The mortality rates remained relatively stable over the years.

The incidence and prevalence rates found with the MYLORD study algorithm are closer to those reported by French network of cancer registries (FRANCIM) than those found with the PALMARO algorithm (regardless of treatment status). The incidence rates are 4.7, 5.15, 4.2 in men and 3.34, 3.53, 2.9 in women for respectively MYLORD, PALMARO, FRANCIM. It highlights the importance of restricting MM identification algorithm based on the SNDS dataset to the treated patients only. SMM does not have an ICD10 code that is separate from MM and may be identified for some to MM by physician in daily practice.

Considering the recent evolution of MM management, such as the increasing number of treatment options and the limited rate of patients with reported LTD (only about 60% of patients), the existing published algorithm was expanded to enable the inclusion of these previously non-identified patients. Indeed, some regimens with lenalidomide and thalidomide do not require any hospitalization for drug administration. Adding the inclusion criteria based on the combination of a treatment regimens received with thalidomide or lenalidomide (at least twice) and two SPEP/UPEP within four months of the year after first dispensing of the drug enabled us to identify a considerable additional number of patients (+ 26%). Actually, in about a quarter of the patients, the first reported MM information was this last inclusion criterion of SPEP/UPEP. An additional feature of the MYLORD algorithm is the accurate identification of incident patients at the time of drug

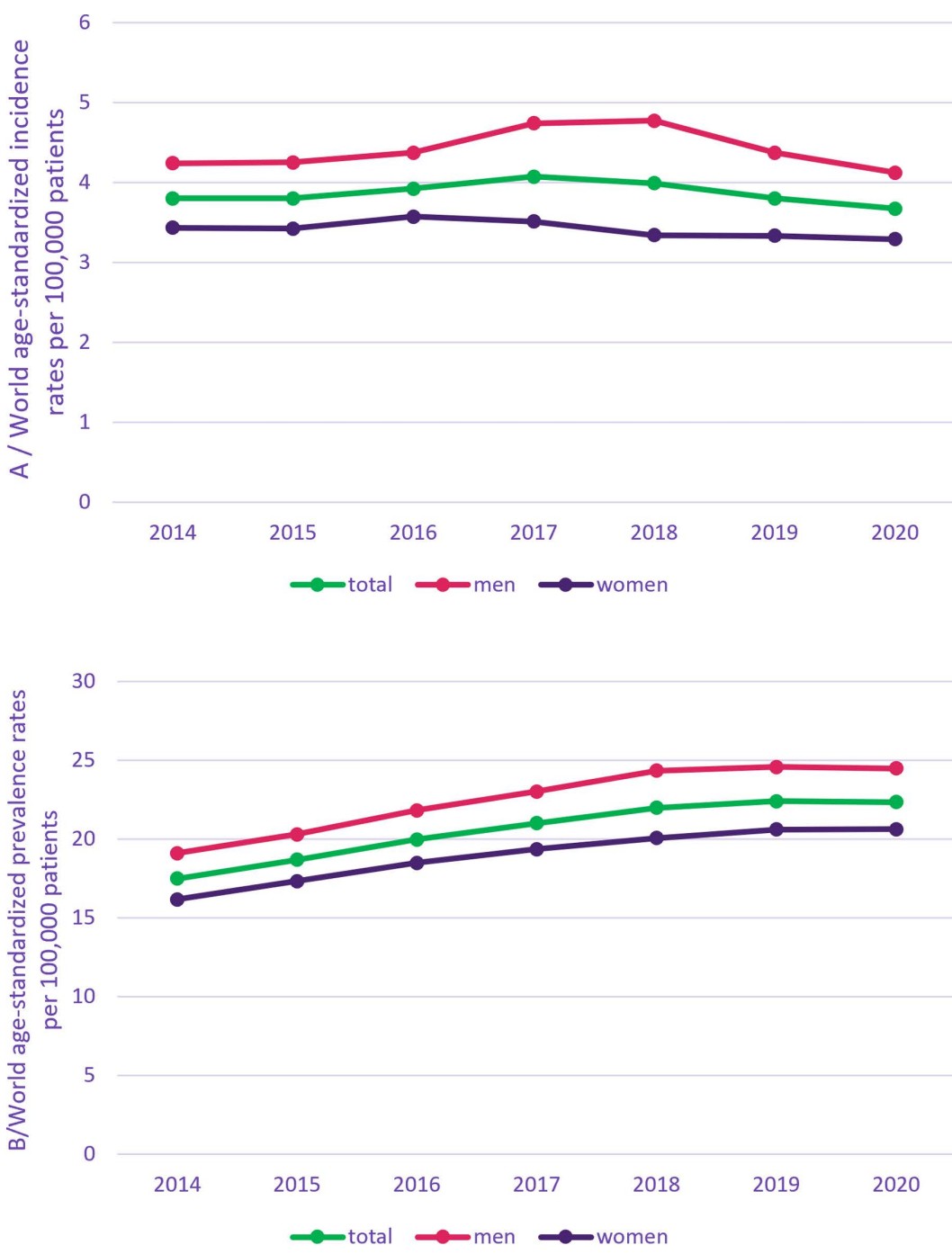

**Fig 2. Standardized incidence rate (A) and standardized prevalence rate (B).**

**Table 3. Prevalence of MM patients in metropolitan France based on the MYLORD algorithm (2014–2020).**

|  | Year | MYLORD |
|---|---|---|
| Number of prevalent patients | 2014 | 22,666 |
|  | 2015 | 24,559 |
|  | 2016 | 26,846 |
|  | 2017 | 28,712 |
|  | 2018 | 30,670 |
|  | 2019 | 31,818 |
|  | 2020 | 32,491 |
| Standardized prevalence rate for 100,000 PY – Men (95%CI) | 2014 | 19.10 (18.68-19.52) |
|  | 2015 | 20.30 (19.87-20.73) |
|  | 2016 | 21.81 (21.37-22.26) |
|  | 2017 | 23.02 (22.57-23.47) |
|  | 2018 | 24.33 (23.87-24.79) |
|  | 2019 | 24.57 (24.12-25.02) |
|  | 2020 | 24.48 (24.04-24.91) |
| Standardized prevalence rate for 100,000 PY – Women (95%CI) | 2014 | 16.17 (15.80-16.54) |
|  | 2015 | 17.32 (16.94-17.70) |
|  | 2016 | 18.49 (18.10-18.88) |
|  | 2017 | 19.35 (18.96-19.75) |
|  | 2018 | 20.07 (19.67-20.47) |
|  | 2019 | 20.61 (20.22-21.01) |
|  | 2020 | 20.62 (20.24-21.01) |

dispensing, whereas the PALMARO algorithm detects MM patients upon hospitalization or at the time of LTD registration, which usually occurs later in the management of MM. Finally, the extension of the identification period to 2006 permitted identification of additional patients.

The MYLORD algorithm is able to provide the French incident and prevalent figures for more recent years, which are not yet available from FRANCIM at the time of this analysis.

This validation study, comparing MYLORD with two external registries, permitted establishment of a robust algorithm for the identification of patients treated for MM. A particular strength of this study is the use of the SNDS as the data source. The SNDS contains individual data used for billing and reimbursement of outpatient healthcare consumption and

**Table 4. Mortality of MM patients based on the MYLORD algorithm (2014–2020).**

|  | Number of deaths in MYLORD study[1] | Standardized mortality rate for 100,000 PY (95%CI) |
|---|---|---|
| 2014 | 2,460 | 1.78 (1.70-1.86) |
| 2015 | 2,728 | 1.93 (1.84-2.01) |
| 2016 | 2,907 | 2.02 (1.94-2.11) |
| 2017 | 2,867 | 1.90 (1.82-1.98) |
| 2018 | 3,063 | 1.97 (1.89-2.05) |
| 2019 | 3,215 | 1.94 (1.86-2.01) |
| 2020 | 3,539 | 1.97 (1.90-2.05) |

[1]No extrapolation to metropolitan French population.

data from the national hospital discharge database, covering 99% of the French population. Thus, owing to the comprehensiveness of coverage of the SNDS, it serves as a suitable tool to estimate epidemiological myeloma indicators.

In addition to the strengths of the SNDS, we acknowledge some limitations of the database. Clinical trial drugs are not fully reported in the database implying a possible underestimation or delayed estimation of incidence. The study was also restricted to patients affiliated with the general scheme. Since the agricultural scheme that covers farmers and agricultural workers and their dependents is not part of the general scheme, this could also lead to an underestimation of the number of MM patients. An increased risk is indeed expected in this population exposed to pesticides.

The absence of detailed clinical outcomes, diagnoses, and treatment indication led to the need for algorithms based on different proxies, such as drug dispensing (not necessarily specific to MM) and electrophoresis tests (specific to MM when associated with the drug dispensing) to identify the patients. In this study, national estimates extrapolated from cancer registries were used to compare data and validate our algorithm.

## Conclusion

We developed an improved algorithm to identify patients treated for myeloma in the SNDS, considering the latest advances in therapeutic management of the patients. The comparison of this algorithm with two other methods has enabled its validation and led to providing robust up to date epidemiological indicators.

The stable incidence and the increase in prevalence from 2014 to 2020 reflect the improvement of the therapeutic management of MM during this period. We recommend using this new algorithm to identify and study MM patients in the SNDS.

## Supporting information

**S1 File.** Supporting information for the article detailed MYLORD algorithm (S1 Appendix), the study flowchart (S1 Fig), the description of Patient characteristics at inclusion in the MYLORD (S1 Table), prevalence rates of multiple myeloma in France from 2014 to 2020 based on MYLORD study population (S2 Table), extrapolated number of prevalent MM patients in France from 2014 to 2020 (S3 Table), incidence rates of multiple myeloma in France from 2014 to 2020 based on MYLORD study population (S4 Table) and extrapolated number of incident MM patients in France from 2014 to 2020 (S5 Table).
(DOCX)

## Acknowledgments

The authors would like to acknowledge the contributions of Florent Daydé (ORCID 0000-0002-9805-5864), Gwendoline Poinsot (0000-0002-5493-3506) and Oana Mihailescu (0000-0001-9991-5654) to the methodology, data management and analysis, and interpretation of results. The authors would also like to thank the Direction de la stratégie, des études et des statistiques (DSES), Département Accès, Traitement et Analyze de la Donnée (DATAD), and Données d'extraction (DEMEX) teams at the Caisse nationale de l'assurance maladie des travailleurs salariés (CNAMTS) for the SNDS data extraction.

## Author contributions

**Conceptualization:** Fanny Raguideau, Ludovic Lamarsalle, Caroline Guilmet, Matthieu Javelot, Marie Pierres.

**Methodology:** Fanny Raguideau, Hélène Denis, Ludovic Lamarsalle.

**Project administration:** Hélène Denis, Marie Pierres.

**Supervision:** Ludovic Lamarsalle, Matthieu Javelot, Marie Pierres.

**Validation:** Cyrille Touzeau, Aurore Perrot.

**Writing – original draft:** Fanny Raguideau.

**Writing – review & editing:** Cyrille Touzeau, Hélène Denis, Caroline Guilmet, Matthieu Javelot, Marie Pierres, Aurore Perrot.

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
