## [Decision Letter · Decision Letter 0]

15 Jan 2025

PONE-D-24-53805Epidemiology of patients treated for myeloma using a new algorithm in the French national health insurance database (SNDS): results from the MYLORD studyPLOS ONE

Dear Dr. Touzeau,

Thank you for submitting your manuscript to PLOS ONE. After careful consideration, we feel that it has merit but does not fully meet PLOS ONE’s publication criteria as it currently stands. Therefore, we invite you to submit a revised version of the manuscript that addresses the points raised during the review process.

I think It is an Important step for the epidemiology of the Multiple Myeloma and could be a pioneer study for the young investigators to inspire.

We look forward to receiving your revised manuscript.

Kind regards,

Mehmet Baysal

Academic Editor

PLOS ONE

Journal Requirements:

“This study was funded by Janssen Cilag France.”

“I have read the journal's policy and the authors of this manuscript have the following competing interests: CT/AP: JANSSEN, Amgen, BMS/Celgene, GSK, Takeda and Sanofi. MJ are MP are employees of Janssen Cilag France. CG was also an employee of Janssen Cilag France at the time this study was conducted. FR, LL, HD are employees of HEVA, who received financial compensation from JANSSEN Cilag France to conduct the study.”

6. PLOS requires an ORCID iD for the corresponding author in Editorial Manager on papers submitted after December 6th, 2016. Please ensure that you have an ORCID iD and that it is validated in Editorial Manager. To do this, go to ‘Update my Information’ (in the upper left-hand corner of the main menu), and click on the Fetch/Validate link next to the ORCID field. This will take you to the ORCID site and allow you to create a new iD or authenticate a pre-existing iD in Editorial Manager.

Reviewers' comments:

Reviewer's Responses to Questions

**Comments to the Author**

1. Is the manuscript technically sound, and do the data support the conclusions?

Reviewer #1: Yes

Reviewer #2: Yes

Reviewer #3: Yes

2. Has the statistical analysis been performed appropriately and rigorously? 

Reviewer #1: Yes

Reviewer #2: Yes

Reviewer #3: Yes

3. Have the authors made all data underlying the findings in their manuscript fully available?

Reviewer #1: Yes

Reviewer #2: Yes

Reviewer #3: Yes

4. Is the manuscript presented in an intelligible fashion and written in standard English?

Reviewer #1: Yes

Reviewer #2: Yes

Reviewer #3: Yes

5. Review Comments to the Author

Reviewer #1: Herein Touzeau et al are comparing two diferrent algorithms in an effort to track accuratelly patients with Multiple Myeloma within the France Health system records. By adding the criteria of >2 perscriptions of lenalidomide or thalidomide together with 2 SPEP or UPEP tests asked by physicians, they identify around 25% more myeloma patients, mostly MM patients that not needed Hospitalization in disease diagnosis Their algorithm the MYLORD algorithm change the incidence and prevelance (age adjusted) of MM in France and this is an important information for clinicians and health services in general.

My concers are

1. The median age of patients detected only with the MYLORD algorithm is lower than those detected with the PALMARO algorithm Based on clinical experience my perception was that outpatient treated MM patients should be older. Is there any explanation?

2. I think that by adding more clinical data comparing patients detected by the MYLORD algorithm and those detected by the PALMARO algorithm will be usefull ( area of living, clinician working place, social status, cycles of yherapy etc)

3. all these outpatient treated MM patients with lenalidomide can we be sure that are not asymptomatic myeloma patients?Having in mind the lenalidomide monotherapy results of two trials in ASM patients is it possible that some ASM were renamed MM in order to receive lena monotherapy?

4. How we exclude the double counts of a patient starting therapy as outpatient one and then needed hospitalization in the upcoming cycles of therapy ?

Reviewer #2: Authors develop an algorithm, MYLORD, to improve identification of treated MM patients, disease incidence and prevalence using national insurance database and compare it with other existing algorithms. Authors claim that inclusion of proxies such as MM drug dispensing and electrophoresis information makes the main difference in identifying patients who are otherwise not identified. A few minor comments below:

-Line 96: Expand the abbreviations SPEP/UPEP

-Line 122: Year 2012 is mentioned for PALMARO study, whereas year 2014 is also mentioned on line 157. Which one is the start year for this study?

-Line 241: Remove extra '.)' symbol after UPEP.

Reviewer #3: MM treatment continuously moves to oral and outpatient settings hampering identication and ultimatively outcome analysis for disease entities. The authors present a new, french system calibrated, approach to generate epidemiologic data incorporating lab and drug usage.

- As this progress, but nor foolproof.

E.g. What quality controlls were instituted to controll the MM diagnosis in pts. identified by LEN use and SPEP alone?

Could also by a routine SPEP in a standard hem-lab profil in a pt on LEN for NHL or MDS?

- the realtively low male preponderance in the frech cohort is a bit at odds with the gobal literature. Any ideas on this fact?

- New alternatives to pt identifiaactio (e.g. use of KI based large laguage modells) should also be discussed

Nice to see the effect of modern MM therapy proven once more

6. PLOS authors have the option to publish the peer review history of their article (what does this mean? ). If published, this will include your full peer review and any attached files.

**Do you want your identity to be public for this peer review?** For information about this choice, including consent withdrawal, please see our Privacy Policy .

Reviewer #1: **Yes: ** Spanoudkis Emmanouil

Reviewer #2: No

Reviewer #3: No

---

## [Author Response · Author response to Decision Letter 1]

5 Mar 2025

Response to reviewer comments

We would like to thank the reviewers for the time taken to review our manuscript and for their insightful comments, as they have led to an improvement of the manuscript. Please see responses to all reviewer comments below.

Reviewer #1:

Herein Touzeau et al are comparing two different algorithms in an effort to track accurately patients with Multiple Myeloma within the France Health system records. By adding the criteria of >2 prescriptions of lenalidomide or thalidomide together with 2 SPEP or UPEP tests asked by physicians, they identify around 25% more myeloma patients, mostly MM patients that not needed Hospitalization in disease diagnosis Their algorithm the MYLORD algorithm change the incidence and prevalence (age adjusted) of MM in France and this is an important information for clinicians and health services in general.

Author response

We are grateful to the reviewer for the effort expended in reviewing our work. Please see below the responses to your individual comments raised.

1. The median age of patients detected only with the MYLORD algorithm is lower than those detected with the PALMARO algorithm Based on clinical experience my perception was that outpatient treated MM patients should be older. Is there any explanation?

Author response

Thank you for raising this comment. When regarding the statistical dispersion of age via usual indicators (mean and standard deviation, median and interquartile range), we can consider that age at diagnosis is quite similar whatever the algorithm used to identify MM patients (see the table below). However, patients identified via the PALMARO algorithm (regardless of treatment status) correspond to all patients with a new diagnosis of MM regardless of treatment status. Thus, SMM patients could have been included via the PALMARO algorithm unlike patients identified via the MYLORD algorithm (only treated patients). This can explain the slight difference on age. This is confirmed by the median age of patients identified via the PALMARO algorithm and treated for MM but also when regarding data from the French cancer registries including all incident cases of MM regardless of treatment status [1] (see the Table below). To clarify, we modify the discussion section (Lines 239 p16 of the version in track changes, to replace previous explanations line 256) :

“The incidence and prevalence rates found with the MYLORD study algorithm are closer to those reported by French network of cancer registries (FRANCIM) than those found with the PALMARO algorithm (regardless of treatment status). The incidence rates are 4.7, 5.15, 4.2 in men and 3.34, 3.53, 2.9 in women for respectively MYLORD, PALMARO, FRANCIM. It highlights the importance of restricting MM identification algorithm based on the SNDS dataset to the treated patients only. SMM does not have an ICD10 code that is separate from MM and may be identified for some to MM by physician in daily practice. “

Also regarding the comparison between MYLORD and PALMARO (treated patients), we do not see older patients for MYLORD because of a bias in PALMARO identification which identifies these patients but later in the disease course. As it has been already pointed out in the manuscript : “An additional feature of the MYLORD algorithm is the accurate identification of incident patients at the time of drug dispensing, whereas the PALMARO algorithm detects MM patients upon hospitalization or at the time of LTD registration, which usually occurs later in the management of MM.”

MYLORD

N=46,202 PALMARO (regardless of treatment status)

N=59,704 PALMARO

(treated patients)

N=36,453 FRANCIM data

(regardless of treatment status)

Mean age (+/- Standard deviation), years 69.54 (+/-12.92) 71.76 (+/-12.50) 70.27 (+/-11.64) NR

Median age, years 71.00 73.00 71.00 74

Q1;Q3 62.00 ; 79.00 64.00 ; 81.00 63.00 ; 79.00 51;89

2. I think that by adding more clinical data comparing patients detected by the MYLORD algorithm and those detected by the PALMARO algorithm will be usefull (area of living, clinician working place, social status, cycles of therapy etc)

Author response

Thank you for this comment questioning the differences between MM patients identified with both algorithms.

The objective of this study was to develop and compare several methodologies to provide robust and updated epidemiological data in France on MM. SNDS data offers the possibility to complete data from cancer registry because of national coverage and good performance of algorithm of detection [2]. For this specific objective, we only presented main characteristics usually presented to describe the epidemiology of the disease (age and gender). The main difference between patients included or not via the two algorithms is treatment status. Thus, covariates associated with the initiation of treatment are mainly clinical data which are lacking or not directly available in the SNDS. We hypothesized that these differences are random regarding other characteristics (area of living or clinician working place) as patients with MM are predominantly cared in public facilities or chartered private non-profit facilities (private practice is very limited in France) and that they may be in some cases related to discontinuous information regarding drug exposure in the database.

3. all these outpatients treated MM patients with lenalidomide can we be sure that are not asymptomatic myeloma patients? Having in mind the lenalidomide monotherapy results of two trials in ASM patients is it possible that some ASM were renamed MM in order to receive lena monotherapy?

Author response

According to European guidelines (ESMO/EHA guidelines, Dimopoulos et al. Hemasphere 2020), treatment of ASM is not recommended (and not approved) in France outside of clinical trials. Drug dispensed during clinical trials are not available in the SNDS (not reimbursed before market authorization). Thus, all patients treated with lenalidomide and without other diseases identified can be considered as MM patients.

4. How we exclude the double counts of a patient starting therapy as outpatient one and then needed hospitalization in the upcoming cycles of therapy ?

Author response

Thank you for this comment raising a lack of clarity in the method section. Patients can be considered incident only once in the study period. Outcomes were analysed by calendar year constituting yearly sub-population. Incident patients were all newly included patients on the considered year of analysis. The index date for these patients was defined as the first date with MM treatment during the study period (2014-2020) corresponding to outpatients therapy or systemic chemotherapy. If patients were still alive followed year and/or treated with a new therapy there were considered as prevalent during this year. We clarified these points in the method section (L104 p7): “The study period was from January 1st, 2014 to December 31th, 2020. A historical period was available since January 1st, 2006. The index date was defined as the first date with MM treatment during the study period (2014-2020) for patients without MM treatment before January 1st, 2014 (incident patients). The index date was January 1st, 2014 for patients with MM treatment before January 1st, 2014 (prevalent patients).” and L113 p8 : “Descriptive statistics were used to characterize the study population. Outcomes were assessed by calendar year constituting yearly sub-populations. Prevalence was defined as all patients eligible in the study who were alive and affiliated to the general scheme on January 1st of the considered year. Incidence was defined as all patients without MM information for 2 years prior to their inclusion. Incident patients were all newly included patients on the considered year of analysis. Patients could be considered incident only once during the study period.”

Reviewer #2:

Authors develop an algorithm, MYLORD, to improve identification of treated MM patients, disease incidence and prevalence using national insurance database and compare it with other existing algorithms. Authors claim that inclusion of proxies such as MM drug dispensing and electrophoresis information makes the main difference in identifying patients who are otherwise not identified. A few minor comments below:

Author response

Thank you for your time and effort reviewing this manuscript. Please see below the responses to your comments which warrant further explanation.

1. Line 96: Expand the abbreviations SPEP/UPEP

Author response

As requested, we expanded the abbreviation.

2. Line 122: Year 2012 is mentioned for PALMARO study, whereas year 2014 is also mentioned on line 157. Which one is the start year for this study?

Author response

Thank you for this comment. The start year of this MYLORD study is 2014. For the PALMARO algorithm, data were available since 2012 in order to have an observational period of two years. To clarify this point, we modified the method section L127 p8 as follow : “As suggested by the Palmaro et al. algorithm, all adults (>18 years) with at least one hospital stay for MM (PD, RD or SAD with ICD-10 C90*) and/or with LTD status for MM (ICD-10 C90*) from January 1st, 2014 to December 31st, 2020 were included, regardless of treatment status (2). Incident patients were patients without MM information within two years prior the first date of MM information during the study period.”

3. Line 241: Remove extra '.)' symbol after UPEP.

Author response

We modified the text as requested. Reviewer #3:

MM treatment continuously moves to oral and outpatient settings hampering identication and ultimatively outcome analysis for disease entities. The authors present a new, french system calibrated, approach to generate epidemiologic data incorporating lab and drug usage.

Author response

Thank you for your time and effort reviewing this manuscript. Please see below the responses to your comments which warrant further explanation.

1. As this progress, but nor foolproof.E.g. What quality controlls were instituted to controll the MM diagnosis in pts. identified by LEN use and SPEP alone? Could also by a routine SPEP in a standard hem-lab profil in a pt on LEN for NHL or MDS?

Author response

Thank you for this comment. First, our algorithm was based on the validated PALMARO algorithm. This algorithm was validated using cancer registry data as gold standard and showed good performance to identify incident MM patients[2]. Regarding patients identified via lenalidomide use, we made sure that patients were not treated for other hematologic diseases by considering the following inclusion/exclusion criteria (see supplemental material):

- all adults treated with lenalidomide or thalidomide (the same drug dispensed at least twice within 90 days)

AND

- with at least 2 Serum Protein Electrophoresis (SPEP) or Urine Protein Electrophoresis (UPEP) within 4 months of the year after first dispensing of the drug in less than 4 months within one year after the first drug dispensation

AND

- without hospital records on the database for indications other than MM were included. Others indications were : myelodysplastic syndrome (PD, RD or SAD with ICD-10 D46*), and/or follicular lymphoma (PD, RD or SAD with ICD-10 C82*) and/or diffuse non-Hodgkin lymphoma (PD, RD or SAD with ICD-10 C83*) and/or peripheral and cutaneous T cell lymphoma (PD, RD or SAD with ICD-10 C84*) and/or other non-Hodgkin lymphoma (PD, RD or SAD with ICD-10 C85*) and/or osteomyelofibrosis (PD, RD or SAD with ICD-10 C47.4) and/or acute panmyelosis with myelofibrosis (PD, RD or SAD with ICD-10 C94.4) and/or POEMS syndrome (PD, RD or SAD with ICD-10 D47.7) and/or amyloidosis (PD, RD or SAD with ICD-10 E85*).

The list of indications excluded were added to the method section L95 P7. For SPEP, this is not a routine test for NHL or MDS. However, it may be performed because of a monoclonal peak related or not to a lymphoma. Thus, we also add this criterion : “at least 2 SPEP/UPEP within 4 months” to be more robust.

2. the relatively low male preponderance in the French cohort is a bit at odds with the gobal literature. Any ideas on this fact?

Author response

Our results showed that 50.2% of the cohorts (PALMARO or MYLORD algorithm) are men. These results are in concordance with French epidemiological data from cancer registries. In 2018, 5,442 new cases of MM were reported in France. Of these, 52% were men [1] . Moreover, our results (see Table 2) depicted higher incidence in men as already showed in the global literature.

3. New alternatives to pt identification (e.g. use of KI based large laguage modells) should also be discussed

Author response

Thank you for this suggestion. As the main objective of this study was to provide update epidemiological data in France, we wanted to work based on the largest database available in France to ensure exhaustivity of the population, this is why our choice was oriented towards using the SNDS. And as explained above, we used an updated algorithm that has already been used on this SNDS database and which was in parallel validated with cancer registry data. Thus, the methodology used here was robust to identify MM patients. It enables to product epidemiological data at national level. Thus, new alternatives such as LLM or AI was not of interest here to identify patients. Moreover, these new tools are not validated.

References

[1] Survie des personnes atteintes de cancer en France métropolitaine (1989-2018) - Les données sur les cancers n.d. https://www.e-cancer.fr/Expertises-et-publications/Les-donnees-sur-les-cancers/Survie-des-personnes-atteintes-de-cancer-en-France-metropolitaine (accessed January 31, 2025).

[2] Palmaro A, Gauthier M, Conte C, Grosclaude P, Despas F, Lapeyre-Mestre M. Identifying multiple myeloma patients using data from the French health insurance databases: Validation using a cancer registry. Medicine (Baltimore) 2017;96:e6189. https://doi.org/10.1097/MD.0000000000006189.

[3] https://static.hevaweb.com/web/PDF/3472fe97b0b06-janssen-mylord-poster-eha2021-lignes-de-traitement.pdf n.d. https://static.hevaweb.com/web/PDF/3472fe97b0b06-janssen-mylord-poster-eha2021-lignes-de-traitement.pdf (accessed January 31, 2025).

---

## [Decision Letter · Decision Letter 1]

24 Mar 2025

Epidemiology of patients treated for multiple myeloma using a new algorithm in the French national health insurance database (SNDS): results from the MYLORD study

PONE-D-24-53805R1

Dear Dr. Touzeau,

We’re pleased to inform you that your manuscript has been judged scientifically suitable for publication and will be formally accepted for publication once it meets all outstanding technical requirements.

Kind regards,

Mehmet Baysal

Academic Editor

PLOS ONE

Additional Editor Comments (optional):

Reviewers' comments:

Reviewer's Responses to Questions

**Comments to the Author**

1. If the authors have adequately addressed your comments raised in a previous round of review and you feel that this manuscript is now acceptable for publication, you may indicate that here to bypass the “Comments to the Author” section, enter your conflict of interest statement in the “Confidential to Editor” section, and submit your "Accept" recommendation.

Reviewer #1: All comments have been addressed

2. Is the manuscript technically sound, and do the data support the conclusions?

Reviewer #1: Yes

3. Has the statistical analysis been performed appropriately and rigorously? 

Reviewer #1: Yes

4. Have the authors made all data underlying the findings in their manuscript fully available?

Reviewer #1: Yes

5. Is the manuscript presented in an intelligible fashion and written in standard English?

Reviewer #1: Yes

6. Review Comments to the Author

Reviewer #1: All querries are well adressed by the authors. I have no further comments My suggestion is Accepted in its present form

7. PLOS authors have the option to publish the peer review history of their article (what does this mean? ). If published, this will include your full peer review and any attached files.

**Do you want your identity to be public for this peer review?** For information about this choice, including consent withdrawal, please see our Privacy Policy .

Reviewer #1: No

---

## [Editor Report · Acceptance letter]

PONE-D-24-53805R1

PLOS ONE

Dear Dr. Touzeau,

I'm pleased to inform you that your manuscript has been deemed suitable for publication in PLOS ONE. Congratulations! Your manuscript is now being handed over to our production team.

Kind regards,

on behalf of

Dr. Mehmet Baysal

Academic Editor

PLOS ONE